# A Target-Agnostic Attack on Deep Models: Exploiting Security Vulnerabilities of Transfer Learning

**Shahbaz Rezaei & Xin Liu**
Department of Computer Science
University of California
Davis, CA 95616, USA
{srezaei,xinliu}@ucdavis.edu

## Abstract

Due to insufficient training data and the high computational cost to train a deep neural network from scratch, transfer learning has been extensively used in many deep-neural-network-based applications. A commonly used transfer learning approach involves taking a part of a pre-trained model, adding a few layers at the end, and re-training the new layers with a small dataset. This approach, while efficient and widely used, imposes a security vulnerability because the pre-trained model used in transfer learning is usually publicly available, including to potential attackers. In this paper, we show that without any additional knowledge other than the pre-trained model, an attacker can launch an effective and efficient brute force attack that can craft instances of input to trigger each target class with high confidence. We assume that the attacker has no access to any target-specific information, including samples from target classes, re-trained model, and probabilities assigned by Softmax to each class, and thus making the attack target-agnostic. These assumptions render all previous attack models inapplicable, to the best of our knowledge. To evaluate the proposed attack, we perform a set of experiments on face recognition and speech recognition tasks and show the effectiveness of the attack. Our work reveals a fundamental security weakness of the Softmax layer when used in transfer learning settings.

## 1 Introduction

Deep learning has been widely used in various applications, such as image classification Parkhi et al. (2015), image segmentation Chen et al. (2016), speech recognition Ji et al. (2018), machine translation Wu et al. (2016), network traffic classification Rezaei & Liu (2019b), etc. Because training a deep model is expensive, time-consuming, and data intensive, it is often undesirable or impractical to train a model from scratch in many applications. In such cases, transfer learning is often adopted to overcome these hurdles.

A typical approach for transfer learning is to transfer a part of the network that has already been trained on a similar task, add one or more layers at the end, and then re-train the model. Since a large part of the model has already been trained on a similar task, the weights are usually kept frozen and only the new layers are trained on the new task. Hence, the number of training parameters is considerably smaller than it is when training the entire model, which allows us to train the model quickly with a small dataset. Transfer learning has been widely used in practice Rezaei & Liu (2019c), including applications such as face recognition Parkhi et al. (2015), text-to-speech synthesis Jia et al. (2018), encrypted traffic classification Rezaei & Liu (2019a), and skin cancer detection Esteva et al. (2017).

One security vulnerability of transfer learning is that pre-trained models, also refereed to as teacher models, are often publicly available. For example, Google Cloud ML tutorial suggests using Google's Inception V3 model as a pre-trained model and Microsoft Cognitive Toolkit (CNTK) suggests using ResNet18 as a pre-trained model for tasks such as flower classification Wang et al.

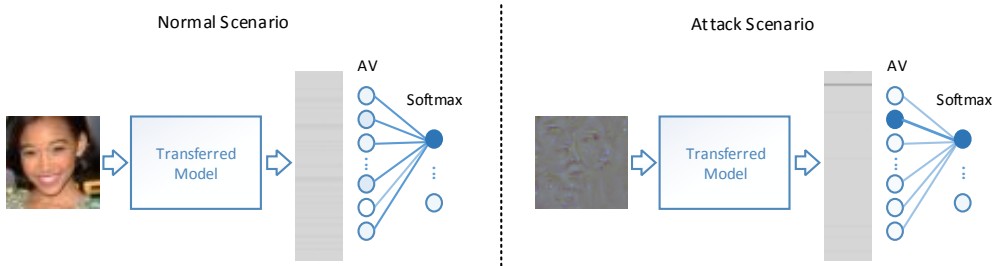

Figure 1: Example of activation vector and how the Softmax layer responses. The image on the left shows the activation vector of a natural face in the training set. The Softmax layer performs Softmax operation over the linear combination of such activation vectors and assigns high confidence to the corresponding class. The target-agnostic image (the image on the right) is crafted such that it activates one neuron in activation vector with extremely large value and all others are almost zero. Such activation vector also fools the softmax layer to produce output with high confidence. Due to the lack of space, we only show the first 400 neurons of the activation vector.

(2018). This means that the part of the model transferred from the pre-trained model is known to potential attackers.

In this paper, we show that an attacker can launch a target-agnostic attack and fool the network when only the pre-trained model is available to the attacker. In our attack, the attacker only knows the pre-trained (teacher) model used to re-train the target (student) model. The attacker does not know the class labels, samples from any target class, the entire re-trained model, or probabilities the model assigns to each class, making it target-agnostic. To the best of our knowledge, these assumptions are more general than those used in any previously proposed attack models, which renders the old models ineffective.

The target-agnostic attack can be adopted in scenarios where fingerprint, face, or voice is used for authentication/verification. In such cases, the attacker usually lacks access to fingerprint or voice samples, which could be used to bypass authentication/verification. Our attack aims to craft an input that triggers any target class with high confidence. The attacker can also continue the crafting process to trigger all target classes. Such adversarial examples can be used to easily bypass authentication/verification systems without having a true sample of the target class. Our work develops a highly effective target-agnostic attack, exploiting the intrinsic characteristic of Softmax in transfer learning settings. Our experiments on face recognition and speech recognition demonstrate the effectiveness of our attack.

In a typical transfer learning procedure, all neural network layers up to the penultimate layer are transferred to a new model and then a Softmax layer is added and re-trained on a new task. We call the scores at the penultimate layer *activation vector* and the part of the model that produces the activation vector *feature extractor*. Hence, the Softmax layer basically computes the softmax operation over the linear combination of activation vector. The left side of Figure 1 presents a natural input and a typical activation vector. Due to the use of linear combination of elements of activation vectors in Softmax layer, not only such patterns can trigger the corresponding classes, but also a large number of other unrelated patterns can also trigger Softmax layer in the same way. In this paper, we show that if we craft an image that produces an activation vector such that one neuron is large and others are almost zero (the right side of Figure 1), it triggers the class for which the weight associated to that neuron is higher in the linear combination. In other words, instead of finding all features that should be activated by feature extractor, we assign very large value to only one neuron to compensate for other neurons that we do not activate.

In summary, the contributions of this paper are as follows:

1. Present a target-agnostic attack in transfer learning settings. We show that if the pre-trained model used during transfer learning is available, an attacker can craft a set of universal

adversarial images that can effectively fool any model re-trained on the pre-trained model. Our attack does not need any training sample from the target model or the target model itself for crafting images. Such a target-agnostic attack has two consequences: I) the crafting time is irrelevant because adversarial images are crafted only once and then they can be used on any model that used the pre-trained model during the transfer learning stage (that is why the attack is called target-agnostic), and II) it does not need to query the target model to craft images. Hence, an attack can craft a set of adversarial images on VGG face model, as an example, and then uses them effectively on any re-trained model based on VGG face.

2. Design a simple approach to exploit the vulnerabilities of Softmax layer. We show that both threshold-based approach, where the model only accept the classification result if the confidence is high, and reject-class-based approach, where the model is trained with an extra class, called reject/null class, to reject adversarial images are prone to our attack.

3. Evaluation of our attack on face recognition and speech recognition tasks. We study the effectiveness of our model in different scenarios and settings.

## 2 RELATED WORK

In general, there are two types of attacks on deep neural networks in literature: I) evasion and 2) data poisoning. In the evasion attack, an attacker aims to craft or modify an input to fool the neural network or force the model to predict a specific target class Elsayed et al. (2018). Various methods have been developed to generate adversarial examples by iteratively modifying pixels in an image using gradient of the loss function with respect to the input to finally fool the network Szegedy et al. (2013); Carlini & Wagner (2017a;b). These attacks usually assume that the gradient of the loss function is available to the attacker. In cases where the gradient is not available, it has been shown than one can still generate adversarial examples if the top 3 (or any other number of) predicted class labels are available Sharif et al. (2016). Interestingly, it has been shown that the adversarial examples are often universal, that is, an adversarial example generated for a model can often fool other models as well Carlini & Wagner (2017a). This allows an attacker to craft adversarial examples from a model she trained and use it on the target model provided that the training set is available.

The second type of attacks on deep neural networks is called data poisoning Shafahi et al. (2018). In the data poisoning attack, an attacker modifies the training dataset to create a backdoor that can be used later to trigger specific neurons which cause mis-classification. In some papers, a specific pattern is generated and added to the training set to fool the network to associate the pattern with a specific target class Sharif et al. (2016); Chen et al. (2017b); Liao et al. (2018); Liu et al. (2017). For instance, these patterns can be an eyeglass in a face recognition task Sharif et al. (2016), randomly chosen patterns Chen et al. (2017b), some specific watermarks or logos Liu et al. (2017), specific patterns to fool malware classifiers Muñoz-González et al. (2017), etc. In some extreme cases, it has been shown that by only modifying a single bit to have a maximum or minimum possible value, one can create a backdoor Alberti et al. (2018). This happens due to the operation of max pooling layer commonly used in convolutional neural networks. After the training phase, the backdoor can be used to fool the network to predict the class label associated with these patterns at inference time.

There are a few studies specifically focused on attacks in transfer learning scenarios Ji et al. (2018); Wang et al. (2018). In Wang et al. (2018), the pre-trained model and an instance of target image are assumed to be available. Assuming that the attacker knows that the first $k$ layers of the pre-trained model copied to the new model, the attacker perturb the source image such that the internal representation (activation vector) of the source image becomes similar to the internal representation of the target image at layer $k$, using pre-trained model. In Ji et al. (2018), first, a set of semantic neighbors are generated for a given source and target input which are used to find the salient features of the source and target class. Then, similar to Wang et al. (2018), the pre-trained feature extractor is used to perturb the source image along the salient features such that their internal representation becomes close. However, these attacks do not work when no instances of the target class is available.

In this paper, we propose a target-agnostic attack on transfer learning. We assume that only the pre-trained model (e.g., VGG face or ResNet18) is available to the attacker. We assume the re-training data and the re-trained model is unknown and not even a single target class sample is available. Our attack model is more general than the previous studies, and thus renders previous attacks on transfer learning infeasible. Note that black-box attacks Sharif et al. (2016); Papernot et al. (2017), where an

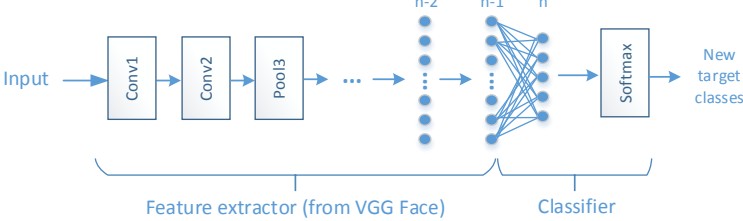

Figure 2: Transfer learning on VGG Face

attacker only have access to the model output, can theoretically be applied in our transfer learning settings. However, a successful black-box attack often needs hundreds to millions of queries to the target model whereas the high effectiveness of our attack means it only needs a few query to the target model to generate adversarial input.

## 3 SYSTEM MODEL

In this paper, we assume that the transferred model trained on a source task is publicly available. This is a reasonable assumption, which in fact is widely used in practice. For instance, Liu et al. (2017) used the VGG face model Parkhi et al. (2015) trained to recognize 2622 identities to recognize 5 new faces. The model is shown in Fig. 2. While our attack targets any transfer-learning-based deep models, we use face recognition based on VGG face as an example for explanation. Fig. 2 shows the typical transfer learning approach for face recognition Parkhi et al. (2015).

In transfer learning, the layers whose weights are transferred to the new model are called ***feature extractor*** that outputs semantic (internal) representation of an input. The last few layers that are re-trained on the new task are called ***classifier***. In typical transfer learning attack scenarios, the transferred model is publicly available, but the re-trained model is not known to an attacker. In other words, the attacker only knows the feature extractor but not the classifier. The previous work on transfer learning Wang et al. (2018); Ji et al. (2018) assumes that at least one sample image from each target class is available because they aim to generate images that produce similar activation vector as the target samples produce. These approaches do not work without samples from the target class.

In this paper, we assume that the attacker does not have access to any samples of the new target classes. Our motivation of the attack is to craft images for models used in systems, such as authentication/verification system, for which there is no target sample available, otherwise the attacker could have just used those samples. In such cases, attackers do not have access to samples of the target classes and, consequently, the previous attacks do not work.

## 4 ATTACK DESIGN

**Design Principle.** To launch an attack with these restrictive assumptions, we need to approach the problem differently. Our attack exploits the key vulnerabilities of the Softmax layer which assigns high confidence labels to vast area of input space that are not necessarily close to the training manifold Gunther et al. (2017). Softmax layer basically performs Softmax operation on the linear combination of activation vector. The activation vector of a real image often shows certain pattern with several triggered neurons, as in Figure 1 (on the left). However, the linear combination of the Softmax layer can also be triggered if only a single neuron in the activation vector has a large value. In other words, each neuron of the activation vector has a direct and linear relation with one or few target classes with different weights. Hence, the attacker can trigger these neurons one by one to see which one is highly associated with each target class.

The main attack idea is to activate the $i^{th}$ neuron at the output of the feature extractor ($n-1^{th}$ layer), denoted by $x_i^{n-1}$, with a high value and keep the other neurons at the same layer zero, similar to the Figure 1 (on the right). After the feature extractor, the model has only a FC layer and a Softmax that outputs the probability of each class. Because of the linear combination used before Softmax

operation, if there exists a neuron at layer $n^{th}$ that associated a large weight to $x_i^{n-1}$, it will become large. Hence, the softmax will assign a high confidence to that class. In order to find an adversary image, we can iteratively try to trigger each neuron at the $(n-1)^{th}$ layer to find an adversary image.

Next, we further explain the attack intuition in more detail using a simple example. Let's assume that the output of feature extractor is layer $(n-1)^{th}$ and we only have two target classes. Let's keep all neurons at layer $(n-1)^{th}$ zero except the $i^{th}$ neuron, denoted by $x_i^{n-1}$. Then, for the last layer, $n^{th}$, we have $x_1^n = W_{1,i}^n x_i^{n-1}$ and $x_2^n = W_{2,i}^n x_i^{n-1}$, and other terms are zero. We omit $b$ for simplicity. Now, if $W_{1,i}^n > W_{2,i}^n$, increasing $x_i^{n-1}$ increases the difference between $x_1^n$ and $x_2^n$. Although the difference increases linearly with $x_i^{n-1}$, the Softmax operation makes the difference exponential. In other words, by increasing $x_i^{n-1}$, one can arbitrarily increase the confidence of the target class whose $W_i^n$ is higher, i.e., class 1 in this example. That is the motivation of the proposed brute force attack.

---

**Algorithm 1** The target-agnostic brute force attack

---

**Input:** M (number of neurons at the output of feature extractor), $I_{img}$ (initial input), $K$ (number of iteration), $F$ (known feature extractor), $\alpha$ (step constant), $T$ (the target model on attack):

1:  **procedure** ATTACK($I_{img}$, $F$, $T$)
2:     **for** i from 1 to $M$ **do**
3:        $Y = 0^m$
4:        $Y[i] = 1000;$                                ▷ Any sufficiently large number
5:        $X = I_{img}$
6:        **for** j from 1 to $K$ **do**
7:            $L = \gamma(F(X)[i]) - Y[i])^2 + \beta(\sum_{l \neq i} relu(F(X)[l] - Y[l])^2)$
8:            $\delta = \frac{\partial L}{\partial x}$
9:            $X = X - \alpha\delta$
10:       **if** $T(X)$ bypasses the authentication **then return** $X$
      **return** $\varnothing$

---

**Algorithm Design.** The brute force algorithm is shown in Algorithm 1. We first iterate through all neurons at the output of the feature extractor and set the target, $Y$, such that at each iteration only one neuron is triggered. We set all elements of $Y$ to zero except for the $i^{th}$ one which can be set to any sufficiently large number, e.g., 1000, in Algorithm 1. Note that $Y$ is a target of the feature extractor, not that of the entire re-trained model. In the case of the VGG face, there are 4096 neurons at this layer. So, we only try 4096 times at maximum. In fact, we will show in the next section that we only need to try a few times to trigger any class and we need way fewer than 4096 attempts to trigger all target classes at least once.

Inside the second loop, we use the derivative of the loss with respect to an input and change the input gradually to decrease the loss. Note that in the loop we only use the pre-trained model and the re-trained target model is not needed. We find that typical MSE loss between Y and feature extractor is very inefficient. For the target activation vector where $i^{th}$ neuron is large and all other neurons are close to zero, the modified loss is defined as follows:

$$L = \gamma(F(x)[i]) - Y[i])^2 + \beta(\sum_{l \neq i} relu(F(x)[l] - Y[l])^2), \tag{1}$$

where $F(X)$ is the output of feature extractor (i.e. activation vector) and $Y$ is the target activation vector. It is similar to the regular MSE loss with two minor changes: I) Because the importance of $i^{th}$ neuron is greater than all other 4095 neurons to our attack, we use $\gamma$ and $\beta$ to control the influence of each part on the loss function. II) Because of the existence of relu function after each fully connected layer to provide non-linearity, any value on the $(-\infty, 0]$ range becomes 0. Hence, instead of crafting an image that has large value in $i^{th}$ neuron and zero value in all other neurons, we aim to craft an image that has large value in $i^{th}$ neuron and any non-positive value in all other neurons. Not only the original MSE might not converge to an adversary example, our revised loss function defined in (1) is much more efficient since the loss function only focuses on neurons that have positive value at each step and ignores the ones that are already negative. This goal is acheived by adding the relu function in the loss function.

**Implication.** We call this type of attack target-agnostic because it does not exploit any information from target's classes, model, or samples. In fact, if the same pre-trained model is used to re-train two different target tasks (models), *A* and *B*, the proposed target-agnostic attack crafts similar adversarial inputs for both *A* and *B* since it only uses the pre-trained model to craft inputs. The implication is that the attacker can craft a set of adversarial inputs with the source model using the proposed attack and use it effectively to attack all re-trained models that use the same pre-trained model. This means that the attack crafting time is not important and one can create a database of likely-to-trigger inputs for each popular pre-trained model, such as the VGG face or ResNet18. Given the simplicity, remarkable effectiveness, and target-agnostic feature of the proposed algorithm, it poses a huge security threat to transfer learning.

## 5 EVALUATION

In this section, we evaluate the effectiveness of our approach using two test cases: Face recognition and speech recognition (Appendix A.2). We use Keras with Tensorflow backend and a server with Intel Xeon W-2155 and Nvidia Titan Xp GPU using Ubuntu 16.04 [1]. We use two metrics to evaluate the proposed attack model: 1) *Number of attempts to break all classes (NABAC):* Assuming that the number of target classes are known, this metric shows how many adversarial input instances are queried, on average, to trigger all target classes at least once with above $99\%$ confidence. 2) *Effectiveness ($X\%$):* This metric shows the ratio of crafted inputs that trigger any target classes with $X\%$ confidence over the total number of crafted inputs. We use $95\%$ and $99\%$ confidence for effectiveness in this paper.

### 5.1 CASE STUDY: FACE RECOGNITION

In this case study, we use the VGG face model Parkhi et al. (2015) as a pre-trained model. We remove the last FC layer and the softmax (SM) layer to make a feature extractor. Then, we pair it with a new FC and SM layer, and re-train the model (while fixing feature extractor) with labeled faces of vision lab at UMass LWF (2016). During re-training, we train the model with Adam optimizer and cross entropy loss function. We set $K = 50000$, $\alpha = 0.1$, $\beta = 0.01$ , and $\gamma = 1$. In some experiments, we add more FC layers before the SM layer, as explained later.

**Number of Target Classes.** Table 1 shows the impact of number of target classes on the attack performance. We use 20 classes with the highest number of samples from UMass dataset LWF (2016). The largest class is George W Bush with 530 samples and the smallest one is Alejandro Toledo with 39 samples. A blank image is used as an intial image. For 5, 10, and 15 classes, we randomly choose a set from 20 classes and re-train and attack the model 50 times and average the results. For 20 classes, we only re-train and attack once. That is the reason we do not show the standard deviation in the table. Table 2 shows the result when we use five images from each class for test set and all other images for training set. Hence, the re-training dataset is imbalanced. To balance the dataset, we undersample all classes to have an equal training size, shown in Table 1.

As it is shown, the effectiveness of the attack on an imbalanced model is higher. However, the NABAC is slightly worse. We find out that on average the weights of SM layer for the class with larger training samples are slightly higher than the other classes. Hence, it is easier to trigger that class with the proposed method which increases the effectiveness. However, it is much harder to trigger the smallest class which makes the NABAC larger. The impact of imbalance re-training dataset is studied in more detail in Appendix A.1, where we show that the probability of triggering a target class directly associated with the number of training samples of that class during re-training. Moreover, the effectiveness and the NABAC improves when the number of target classes decreases, as expected. Note that in all scenarios, the effectiveness is greater than $75\%$. It means that the first crafted image has more than $75\%$ chance of bypassing the authentication system (or any other application). It basically means that the traditional approach of limiting the number of queries to prevent brute-force-based attack does not work here.

**Number of Layers to Re-train.** In previous experiments, we assume that the weights of the feature extractor transferred from the pre-trained model are fixed during re-training and only the last FC layer is changed. One can tune more layers during re-training. Fig. 3(a) shows the impact of tuning

---

[1]The implementation is available in https://github.com/shrezaei/Target-Agnostic-Attack

Table 1: Attack performance on balanced re-training dataset. *Acc*, *NABAC*, and *Eff* stands for accuracy, number of attempts to break all classes, and effectiveness, respectively.

| Target classes | Balanced dataset | | | |
| --- | --- | --- | --- | --- |
| | Acc | NABAC | Eff(95%) | Eff(99%) |
| 5 | 99.12% ± .27 | 48.25 ± 42.5 | 91.68% ± 5.69 | 87.82% ± 6.98 |
| 10 | 98.43% ± .23 | 149.97 ± 132.15 | 88.87% ± 2.46 | 83.07% ± 3.31 |
| 15 | 97.16% ± 1.64 | 323.36 ± 253.56 | 87.79% ± 2.42 | 82.05% ± 2.74 |
| 20 | 96.87% | 413 | 87.17% | 79.16% |

Table 2: Attack performance on imbalanced re-training dataset. *Acc*, *NABAC*, and *Eff* stands for accuracy, number of attempts to break all classes, and effectiveness, respectively.

| Target classes | Imbalanced dataset | | | |
| --- | --- | --- | --- | --- |
| | Acc | NABAC | Eff(95%) | Eff(99%) |
| 5 | 99.21% ± .29 | 63.29 ± 80.30 | 93.52% ± 5.07 | 90.23% ± 5.71 |
| 10 | 98.47% ± .81 | 264.80 ± 111.09 | 91.14% ± 3.65 | 86.28% ± 5.40 |
| 15 | 98.01% ± 1.39 | 451.45 ± 244.31 | 90.41% ± 1.89 | 85.31% ± 2.48 |
| 20 | 97.07% | 2836 | 88.72% | 82.93% |

more layers on the effectiveness and accuracy. Note that we assume that attacker does not know anything about the target model. Hence, in this experiment, the attacker still uses the pre-trained feature extractor up until the last FC layer. That means the pre-trained feature extractor that the attacker uses is slightly different from the re-trained model. In Fig. 3(a), X axis represents the layer from which we start tuning up to the last FC layer. Due to the small re-training dataset, as the number of tuning layers increases the accuracy drops. However, by tuning more layers, the pre-trained model that the attacker has access to becomes more different from the re-trained model. That is why the effectiveness of the attack decreases. Similarly, NABAC increases, as shown in Fig. 3(b). Despite the difference between the re-trained model and the model the attacker has access to, the attack is still effective, which means that the pre-trained model cannot be changed dramatically during re-training process and re-training more layers is not an effective defense strategy.

**Number of New Layers in the Re-trained Model.** Next, we measure how adding and training more layers (pair of FC + Relu) after feature extractor can affect the proposed attack effectiveness. In this experiment, we use 5 balanced target classes. As shown in Table 3, adding more layers decreases the accuracy of the re-trained model because the re-training dataset is small and not enough to train more layers from scratch. The effectiveness of the attack decreases sightly as more new layers are tuned. When adding more new layers, not all target classes are affected equally and some classes may become harder to trigger. That is why NABAC increases. The goal of our attack is to have an activation vector with only one large value. However, each extra layer, added after feature extractor,

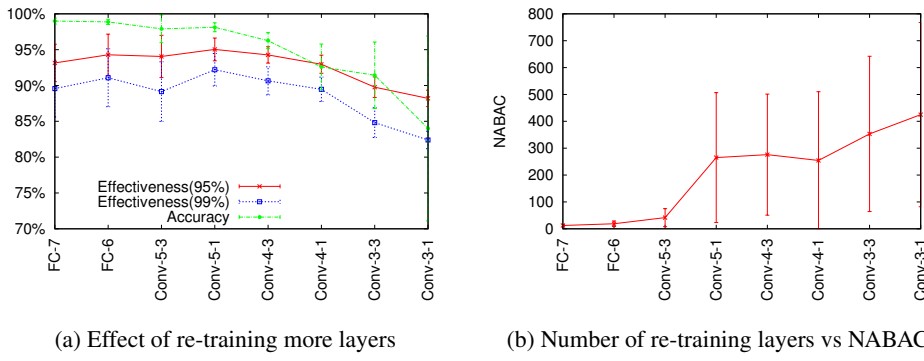

(a) Effect of re-training more layers        (b) Number of re-training layers vs NABAC

Figure 3: Effect of number of re-training layers

Table 3: Effect of number of new layers in the re-trained model

| # of new layers | Accuracy | NABAC | Effectiveness(95%) | Effectiveness(99%) |
|---|---|---|---|---|
| 1 | $99.12\% \pm .27$ | $48.25 \pm 42.5$ | $91.68\% \pm 5.69$ | $87.82\% \pm 6.98$ |
| 2 | $98.24\% \pm 2.10$ | $51.87 \pm 39.94$ | $91.57\% \pm 4.87$ | $86.45\% \pm 5.35$ |
| 3 | $95.46\% \pm 4.2$ | $257.26 \pm 387.16$ | $89.45\% \pm 8.20$ | $85.67\% \pm 8.88$ |

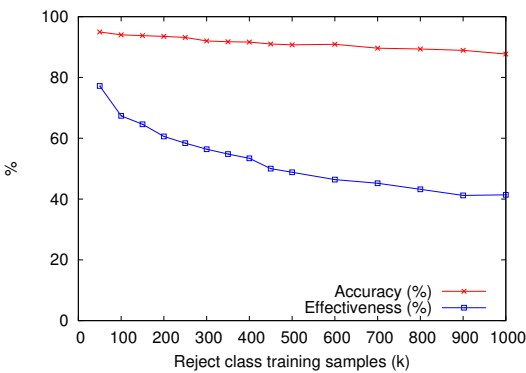

Figure 4: Number of reject training samples vs effectiveness/accuracy

smooths out the single large value and distributes it to more neurons in activation vector. That is the reason the attack becomes less effective when more new layers are added.

**Attack Effectiveness on A Classifier with Reject Class.** It has been shown that relying on a threshold to reject or accept the classification result is not accurate because for the vast space of unknown inputs Softmax provides high confidence scores Nguyen et al. (2015). Hence, we add an extra class to the softmax layer, similar to Hosseini et al. (2017), called reject/unknown class. During re-training, we choose random sample images from entire UMass dataset, except the classes that we choose for target faces, and label them as reject class. We vary the number of training samples for reject class to see its effect on accuracy and effectiveness. Figure 4 illustrates the trade-off between the accuracy of the re-trained model and the effectiveness of our attack. The lowest effectiveness, for which the accuracy is $87.72\%$, is $41.40\%$ which is still high. Hence, the classifier with a reject class option is still prone to our target-agnostic attack.

**Attack Comparison with black-box attack and baseline attack.** We compare our attack with a black-box attack and a baseline. For the baseline attack, we choose random face images from UMass dataset that have not been used for training. Interestingly, for the threshold-based model, there is a $12.60\%$ chance that a random face image triggers an output class with high probability, as shown in Table 4. A model with a reject class can effectively prevent baseline attack since none of the random images fool the model. Moreover, we use Zoo black-box attack Chen et al. (2017a) as comparison. The default configuration of untargeted Zoo attack yields a very low effectiveness of $18.20\%$. After hyper-parameter tuning and setting the confidence of the crafted images in the algorithm to $0.99\%$, Zoo achieves its highest effectiveness of $76.12\%$. The effectiveness of all attacks are higher against the threshold-based model than the model with a reject class. Our attack needs only one query to the re-trained (student) model because it crafts the images using the publicly available pre-trained (teacher) model. Any black-box attack, such as Zoo, that depends only on the student models needs a significant number of queries which is easy to defend by limiting the number of queries.

## 6 DISCUSSION

In this paper, we show that the public information from transfer learning settings can be exploited to fool Softmax-based classifier. The main vulnerabilities of the Softmax layer comes from the fact that it assigns a high confidence output to inputs that are far away from the training input distribution. This drawback has been shown in studies that investigate open-set problem Bendale & Boult (2016); Gunther et al. (2017). In other words, Softmax-based models are vulnerable to inputs with different

Table 4: Attack Comparison. *NQT*, *NQS*, and *Ef* stands for number of query to the teacher model, number of query to the student model, and effectiveness, respectively.

| Attack | Threshold-based model | | | With reject class | | |
|---|---|---|---|---|---|---|
| type | NQT | NQS | Ef(99%) | NQT | NQS | Ef |
| Our attack | 50,000 | 1 | 87.82% | 50,000 | 1 | 78.24% |
| Zoo (black-box) | - | 1,036,800 | 76.12% | - | 816,800 | 81.01% |
| Baseline (random) | - | 1 | 12.60% | - | 1 | 00.00% |

distribution than their training set. To mitigate the problem and defeat our attack, we use a recent novel classifier for the open-set problem, called extreme value machine (EVM) Rudd et al. (2017), that aims to fit a distribution to the activation vector (Figure 1) rather than a linear combination based on Softmax operation. We follow the experimental setting similar to Gunther et al. (2017). The accuracy of the EVM-based model is $95.60\%$, which is lower than the softmax-based model in our experiments, and the model successfully defeat all our crafted images. The main reason that EVM can be used as a defense mechanism is that the activation vector of our crafted images are far from the activation vector of any image in the training set. However, EVM has its own vulnerability: we find out that by feeding images of random faces (UMass dataset in our study), there is a $7.38\%$ chance that the EVM-base model classifies the input as one of the target classes. Hence, more robust model is needed to defend our attack and also work well in open-set scenarios.

Another approach to defend the vulnerability of the Softmax layer is to check all elements of activation vector and avoid classification of suspicious inputs. In other words, if an activation element is significantly larger than what it should normally be, we can label the input image as malicious. In our experiment, we find that the average value of the largest neurons in activation vector is around 23.86 for normal face images and the largest value we observed is 47.22. So, if we define a threshold for the maximum value in activation vector to be around 50, it is possible to detect crafted images with our attack. In our attack scenario, we craft inputs with the maximum value in activation vector of 1000, which is easily detectable, if checked. We perform an experiment to see if our attack work when this value is much smaller and in a normal range. By crafting images with max value in activation vector of 50, instead of 1000, the effectiveness of our attack is dramatically reduced ( $0.0007\%$). However, this threshold may lead to a large false positive in inference time. With the max value of 100 and 200, the effectiveness is $0.058\%$ and $0.27\%$, respectively. Hence, if the threshold value for anomaly detection is chosen meticulously, it can serve as a defense mechanism for our attack with the cost of increasing false positive (labeling some natural face images as malicious).

## 7 CONCLUSION

In this paper, we develop an efficient brute force attack on transfer learning for deep neural networks - the attack exploits a fundamental vulnerability of the Softmax layer that can be easily exploited when transfer learning is used. We assume that the attacker only knows the transferred model and its weights, and does not have access to the re-trained model, the re-trained dataset, and the re-trained model's output. Our evaluations based on face recognition and speech recognition show that with a handful of attempts, the attacker can craft adversarial samples that can trigger all classes despite the fact that the attacker does not know the re-trained model and model's target classes. The target-agnostic feature of the attack allows the attacker to use the same set of crafted images for different re-trained models and achieve high effectiveness when the models use the same pre-trained model. The proposed target-agnostic attack reveals a fundamental challenge of Softmax layer in transfer learning settings: because the Softmax layer assign high confidence output to vast space of unseen inputs, a simple brute-force attack can operate surprisingly effective. To defeat the target-agnostic attack, the model should consider the distribution of the activation vector, like EVM method, not the linear combination alone, like Softmax layer. Nevertheless, there is a fundamental trade-off between accuracy and robustness and it should be tuned based on the sensitivity of the application.

### ACKNOWLEDGMENTS

This work was supported by the National Science Foundation (NSF) under Grant CNS-1547461, Grant CNS-1718901, and Grant IIS-1838207.

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

## A  APPENDIX

### A.1  CASE STUDY: FACE RECOGNITION

**Choice of Initial Image.** To generate adversarial images using Algorithm 1, we need to start with an initial image. To find out whether the initial image we start with has any impact on the brute force attack, we conduct 3 different experiments. We use random input, blank image (with all pixel set to one), and random images of celebrities. The results are shown in Fig. 5. First column shows crafted images starting from the random input. Second column illustrates crafted images from blank image.

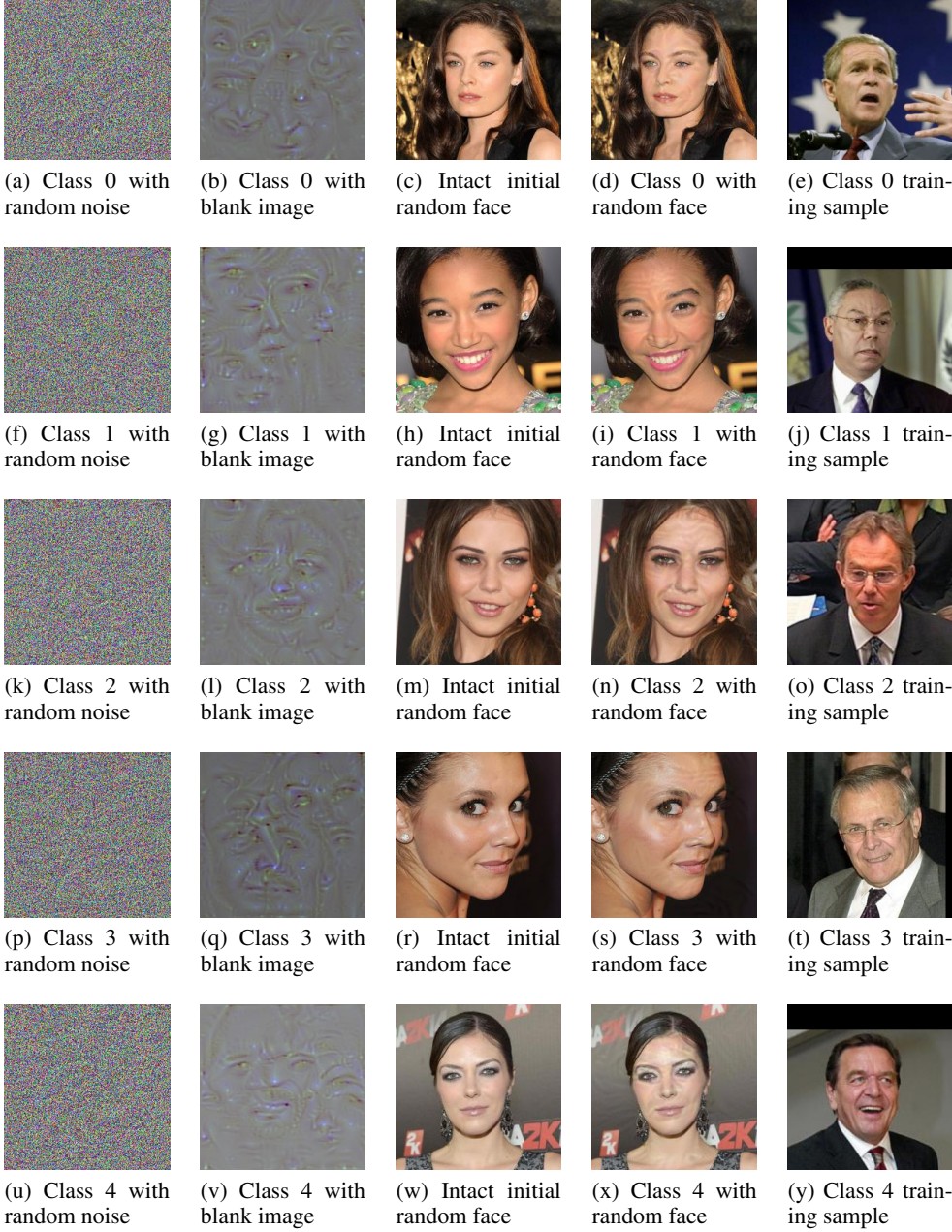

Figure 5: First column shows crafted images starting from the random input. Second column illustrates crafted images from blank image. Third and fourth columns show the initial images and the crafted images from the initial image, respectively. The fifth column illustrates a sample image from each class that is used for re-training.

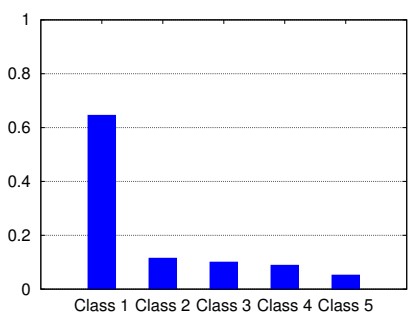

(a) A typical target class distribution

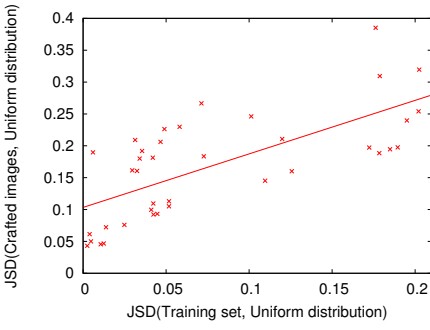

(b) Training dataset vs target class distribution

Figure 6: Target class distribution

Third and fourth columns show the initial images and the crafted images from the initial image, respectively. The fifth column illustrates a sample image from each class that is used for re-training. In our experiment, the choice of initial image has negligible impact on effectiveness of our attack.

Table 5 shows the result of using different initial images on the attack performance. We only re-train a model once with 5 randomly chosen faces and we achieve 99.38% accuracy. Then, we launch the attack on the same model 3 times, each with a different initial image. Although using a face image marginally improves the attack performance, the impact is negligible and the other initial input cases are still considerably effective.

Table 5: Impact of initial input on the attack

| Initial input | NABAC | Effectiveness(95%) | Effectiveness(99%) |
|---|---|---|---|
| Blank | 18 | 98.37% | 98.37% |
| Random | 19 | 98.37% | 97.22% |
| A face image | 18 | 99.83% | 99.19% |

**Distribution of Target Classes.** Fig. 6(a) illustrates a typical distribution of target classes triggered by crafted images of the proposed method. It is clear that the distribution is far from Uniform. It basically means that more neurons in layer $n-1$ are associated with class 1 and, hence, during brute force attack, more crafted images will trigger that class.

To measure the impact of re-training set on the distribution of target classes, we use Jensen-Shannon distance (JSD). Jensen-Shannon divergence measures the similarity between two distributions as follows:

$$JSD(P||Q) = \frac{1}{2}D(P||M) + \frac{1}{2}D(Q||M) \tag{2}$$

where D(.) is Kullback-Leibler divergence and $M = \frac{1}{2}(P + Q)$. Square root of JSD is a metric that we use to compare the similarity between the distribution of data samples in re-training dataset versus the distribution of triggered classes with adversarial inputs of our method.

We find that distribution of training samples during re-training can affect the target class distribution. Fig. 6(b) shows the JS distance of training set distribution and Uniform distribution versus JS distance of target class distribution and Uniform distribution. For each data point, we pick 5 random persons from UMass dataset and then re-train the VGG face model with. The line in Fig. 6(b) represents the linear regression of all data point. The figure shows that when the training set of re-training phase becomes more non-Uniform, the target class distribution becomes even more non-Uniform.

A.2 CASE STUDY: SPEECH RECOGNITION

In Ji et al. (2018), a speech recognition model for digits were re-trained to detect speech commands. Following the same experiment, a model first pre-trained on the Pannous Speech dataset dig (2017) containing utterance of ten digits. Then, we randomly pick 5 classes from speech command dataset

Table 6: Effect of number of target classes on the proposed attack

| # of target classes | Accuracy | NABAC | Effectiveness(95%) | Effectiveness(99%) |
|---|---|---|---|---|
| 5 | 97.38% | 37 | 100.00% | 98.21% |
| 10 | 93.30% | 114 | 95.80% | 93.75% |
| 15 | 85.72% | 812 | 92.22% | 84.17% |

Table 7: Effect of re-training set size

| # of samples per class | Accuracy | NABAC | Effectiveness(95%) | Effectiveness(99%) |
|---|---|---|---|---|
| 50 | 77.56% | 13 | 97.48% | 95.00% |
| 100 | 82.46% | 17 | 97.21% | 95.23% |
| 200 | 85.51% | 21 | 98.25% | 96.82% |
| 1000 | 89.89% | 17 | 98.60% | 97.64% |
| 2000 | 92.04% | 17 | 98.60% | 97.81% |

com (2017) to re-train the model. $80\%$ of the dataset is used for fine-tuning and $20\%$ for inference. Due to the lack of space and similarity of the results with previous case study, we omit most experiments with similar results. We use a 2D CNN model with 3 building block, each of which contains convolutional layers, Relu activation, and pooling layer, followed by 2 FC layers and softmax layer at the end. The input is the Mel-Frequency Cepstral Coefficients (MFCC) of the wave files. Similar to the previous case study, we replace the SM layer and re-train the model by only tuning the last FC and SM layer.

**Number of Target Classes.** Table 6 shows the impact of number of target classes on the accuracy of the model and attack performance. Similar to the face recognition experiment, we start with a blank input (a 2D MFCC with 0 for all elements) and we use 70 and $0.1$ for $k$ and $\alpha$, respectively. As expected, the accuracy drops when the number of target classes increases. Since ten classes representing digits exist in both the pre-training dataset (Pannous dig (2017)) and the re-training dataset (speech command com (2017)), these classes are much easier for the target model to re-train with high accuracy in comparison with other classes, such as stop or left command. Hence, the re-trained model has more neuron connections to help classify digit classes which makes it harder for both the model to classify the other classes and the proposed attack to craft adversarial input for the non-digit classes. That is why we observe more dramatic decrease in accuracy and attack performance when the number of target classes increases.

**Re-training Sample Size.** Unlike face recognition case study in which most re-training classes have fewer than 100 samples, speech command dataset com (2017) contains more than 2000 samples for each class. Hence, we conduct an experiment to study the effect of re-training sample size on model and attack performance. We choose six classes (commands) that the pre-trained model did not trained on, i.e., left, right, down, up, go, and stop speech commands. Table 7 shows the impact of re-training set size on the model and attack performance. As expected, increasing the re-training set size improves the accuracy of the model. However, the accuracy of the re-trained model and the re-training set size have a negligible effect on the performance of proposed attack. By comparing Table 6 and Table 7, we realize that the attack performance is directly affected by the number of target classes, but it is not significantly affected by the accuracy of the re-trained model.

