# OpenReview forum: "A Target-Agnostic Attack on Deep Models: Exploiting Security Vulnerabilities of Transfer Learning"
_ICLR.cc/2020/Conference — Accept (Poster)_

### Official Review · AnonReviewer3 · 2019-10-17
**Official Blind Review #3**

**Rating:** 6

**Review:**

This paper proposes an interesting new adversarial attack concern: any machine learning that use a linear classifier on an off-the-shelf feature extractor, suffers the risk that the feature extractor has adversarial examples that output arbitrary feature vectors. The paper proposes a concrete attack where a set of samples is crafted to activate each feature vector.


Pro:

The setup is interesting and seems novel. To my knowledge, attack on the features used in transfer learning has not been considered in the literature. Several adversarial attack papers consider attacking the network layer by layer, or attack an intermediate layer, but they are different from the proposed setup in this paper.

The proposed attack is simple but effective (though unsurprisingly).

The experiments show reasonable performance even beyond the motivating setup, for example, with fine-tuning or with non-linear classifiers on the feature space.

Con:

I think the range of the input feature (to the linear classifier) should be reported. A simple check on the range of the input feature should be a good defense. Therefore, the input feature should not be unreasonably large.

I think the experiments need comparison with the naive baseline by simply trying random faces. It seems that without any attack, it should be possible to try random faces and find one that is incorrectly classified. Assuming the model it well calibrated, among the time it says it is 99% confident, it should still make a mistake with approximately 1% probability. In addition, deep networks are known to be not calibrated and over confident.

I do not think the attack is task-agnostic, because the attacker still needs to query the final linear layer as a black box to find which feature unit to activate. This is a special hybrid setup, where the feature layer is white-box while the final layers are black box. I think task-agnostic is somewhat an oversell.


Overall I believe this paper is okay. It explores a novel task with a somewhat interesting approach, with sufficient empirical support.



**Experience Assessment:**

I have published one or two papers in this area.

**Review Assessment: Checking Correctness Of Derivations And Theory:**

I assessed the sensibility of the derivations and theory.

**Review Assessment: Checking Correctness Of Experiments:**

I assessed the sensibility of the experiments.

**Review Assessment: Thoroughness In Paper Reading:**

I read the paper at least twice and used my best judgement in assessing the paper.

---

> ### Author Response · Authors · 2019-11-08
> **Response to the comments**
>
> 1. Thanks for your great suggestion. This approach can be used as a defense mechanism. In our experiments, we find that the value of large neuron in activation vector is around 23.86 for normal face images and the largest value we observed was 47.22. So, if we define a threshold for maximum value in activation vector to be around 50, it is possible to detect crafted images with our attack. In our attack scenario, we craft inputs with maximum value in activation vector of 1000, which is easily detectable, if checked. After seeing your suggestion, we performed an experiment to see if our attack work when this value if much smaller and in a normal range. By crafting images with max value in activation vector of 50, instead of 1000, the effectiveness of our attack is dramatically reduced (~0.0007%). With the max value of 100 and 200, the effectiveness is 0.058% and 0.27%, respectively. Hence, if the threshold value for anomaly detection is chosen meticulously, it can serve as a defense mechanism for our attack. Should the paper be accepted, we will include this discussion as a defense strategy.
>
> 2. Thanks for your suggestion. We basically perform two scenarios. One with reject class and one without reject class. When reject class is not used, we rely on confidence to classify (authenticate in this case). For 95% confidence scenario, if random face images are fed to the model, there is a 27.80% chance of success. For 99% confidence scenario, the chance of success is 12.60%. These values are considerably lower than our attack, but still high enough to make an attack based on random face images possible. However, for models with a reject class, we find that none of the random images triggers the model (0.0% success rate). If the paper is accepted, we will cover this in the result section.
>
> 3. We did not mention that our model is task-agnostic. We only mentioned that it is target-agnostic, although it is, in fact, task-agnostic. To clarify, our model is target agnostic, because it does not need any sample from the target class it triggers. Moreover, it is indeed task-agnostic in the sense that the crafted images do not depend on target tasks. For example, let’s assume we have used VGG Face for two different tasks: face recognition, and facial expression recognition. These two tasks use the VGG Face in a transfer learning setting with two different datasets for re-training. However, the images that our attack crafts are exactly the same for both tasks. The reason is that during crafting, we only use the teacher model (VGG Face) to craft images which is always VGG Face regardless of the target task. Then, we test these images on the target task to see if they work. Hence, our crafted images only depend on the source task and the teacher model.

---

### Official Review · AnonReviewer2 · 2019-10-20
**Official Blind Review #2**

**Rating:** 6

**Review:**

In this paper, the authors proposed an attack scheme to any model that pretrained from a general model. The merit comes from that the attacker, by taking advantage of the vulnerability of softmax, has no access to examples from the target task to finetune the model.

Pros:
-	The setting where the authors focused on is much more practical, i.e., the attacker is blind to examples in a target task.
-	The blackdoor for attack in this work, namely the softmax layer, is novel and interesting to me, at least to my knowledge.
-	The paper is well written and easy to follow.

Cons:
-	In the experiments, the authors should still compare one or two black-box attacks which also use model outputs only, to demonstrate the effectiveness/efficiency of the proposed attack scheme.
-	Considering the design of \gamma and \beta in Eqn. (1), it is expected to investigate the influence of both on the performance of the attack, as well as the vanilla version of Eqn. (1) by only considering MSE.
-	Currently, it seems that the authors did not study the influence of different target datasets to finetune. It is highly expected that the same adversarial input crafted using Alg. 1 could fool all networks finetuned using different datasets. What if a wildly different dataset is used to finetune the pretrained model?

**Experience Assessment:**

I do not know much about this area.

**Review Assessment: Checking Correctness Of Derivations And Theory:**

I carefully checked the derivations and theory.

**Review Assessment: Checking Correctness Of Experiments:**

I carefully checked the experiments.

**Review Assessment: Thoroughness In Paper Reading:**

I read the paper at least twice and used my best judgement in assessing the paper.

---

> ### Author Response · Authors · 2019-11-08
> **Response to the comments**
>
> 1. Should the paper be accepted, we will add the comparison to black-box attacks. Note that typical black-box attacks, such Sharif et al. (2016) or Papernot et al. (2017), needs tens to hundreds of queries, if not thousands and more, which is considerably larger than our method and can be easily defended by setting a threshold. We want to emphasize that the number of queries in black-box setting is different from “Number of Attempts” that is used in our paper. (We will revise “Number of attempts” to “Number of attempts to break all classes” to make it more clear.)  Number of attempts shows the number queries to the target model to trigger all classes at least once. It is used in the paper to show that sometimes one class is difficult  to trigger and we need many attempts to trigger that class. But, even in those cases, the number of queries to trigger any class (except the reject class) is astonishingly low. It can be obtained by observing effectiveness. For example, if the effectiveness is 50%, it means that on average you need to query the target model twice to trigger any class which is considerably lower than any black-box attack. We will release comparison results in the next few days or in the revised version. Note that we do not use the target model to craft images. We only use the public teacher model to craft images. Hence, we only query the target model once after the image is crafted, and if it does not work, we craft another image from scratch by triggering another neuron in the activation vector.
>
> 2. In our experiments, we find that the result is not very sensitive to the value of gamma and beta if the order difference is in range of hundreds to tens of thousands. It only slightly affects the speed of convergence. But if we use the same value for beta and gamma, the optimization does not converge because there is only one term that is multiplied by gamma, but thousands of terms are multiplied by beta. So, the optimization mainly focuses on the second term, and consequently it hardly tries to trigger the neuron in the gamma term with a high value, which is the core of our attack. Considering only the MSE part also suffers from the same problem. The existence of the relu function in our loss function is also important. Basically, if the value before applying relu is -10 or -10000, the result would be zero in either case due to the existence of relu in the model. Since our goal is to make all but one neurons to have a zero value, we put the relu to make sure the optimization process does not try to waste time on crafting an input that triggers the neurons before relu with the exact value of zero. As long as the value before relu is zero or negative, that is fine. Hence, it dramatically speeds up the convergence of the optimization. Without it, the optimization problem takes a very long time to craft a good image, or even diverge.
>
> 3. In our experiments, we basically use UMass LWF for re-training in the following manner: in the experiments with N faces, we randomly pick N faces from the UMass dataset that have at least 30 images. Then, we re-train the model and launch the attack. We do that several times with different sets of faces from the UMass dataset and the result was consistent. We are not sure if the UMass dataset is considered widely different from the VGG face dataset. However, the transfer learning procedure that we launch our attack on works only when the source and target tasks are related and share similarities. So that is the best we can think of. We will be happy to try additional datasets if the reviewer has any suggestions.

---

> ### Author Response · Authors · 2019-11-15
> **Comparison with Black-box attacks**
>
> 3. We compare our attack with a black-box attack, called ZOO, in [A], which the source code is available at https://github.com/IBM/ZOO-Attack. We use the same settings as in Section 4.2 of [A] with one exception. We do not stop the optimization process when the target class probability becomes higher than other classes because in our attack scenario, instead, we assume that only images with higher than 95% confidence in one of the classes are accepted. Otherwise, the input is rejected. So, our attack setting is more restricted and that is the reason their black-box attack performs worse in our setting than in their paper. We choose a random image and a random target for the attack. We use the re-trained model on 5 faces as the student model and VGGFace as the teacher model. We set the hard limit of 5,000,000 queries per image. The average query is only computed for successful attacks. In our experiment, the successful rate is 18.2% (in other words, 4 out of 5 attacks fail after 5m attempts). Among the successful ones, Zoo attack needs 1,036,800 queries to the black-box student model on average. In comparison, our attack needs 50,000 queries to the teacher model, 1 query to the black-box student model, and the successful rate (effectiveness) is 91.68%. We need to also emphasize that the number of queries to the teacher model is not critical because the teacher model is publically available and we can download and query the teacher model on a sever of the attacker’s. Additionally, since our attack is target-agnostic, the crafted images only depend on the teacher model. Hence, one can craft a set of potential adversarial images and use it over and over again on multiple student models. It means that, for example, once the crafted images are available, an attacker can use them without a need to query the teacher model. In other words, the number of query is essentially 1 to the student model with an effectiveness of 91.68%. The advantage of using the teacher model gives us enough advantage that no black-box attack without the teacher model can beat us.
>
> Given the time limitation, this is the only model we can implement. We think it clearly demonstrates the difference between our model and a black-box model. If reviewers have specific suggestions on black-box models, we are happy to compare with them further.
>
> [A] Chen, P. Y., Zhang, H., Sharma, Y., Yi, J., & Hsieh, C. J. (2017, November). Zoo: Zeroth order optimization based black-box attacks to deep neural networks without training substitute models. In Proceedings of the 10th ACM Workshop on Artificial Intelligence and Security (pp. 15-26). ACM.

---

### Official Review · AnonReviewer1 · 2019-10-22
**Official Blind Review #1**

**Rating:** 6

**Review:**

The paper proposed a method of attacking deep neural networks that were trained using transfer learning. The primary claim is that the proposed technique only requires knowledge of the base model (i.e., if the frozen parameters taken from pretrained VGG, ResNET models are known, then that is sufficient) and doesn’t require any samples of target classes of the post-transfer application to successfully attack the DNN.

Core idea is that Softmax layers used in classification can be exploited to craft a perturbation that will fool the network. The algorithm uses a brute force approach to iterate through each neuron on the final layer before the softmax. For each activated neuron, all other neurons are zeroed out and a regularized loss is computed between the activated feature and the neuron. The gradient of the loss function is then used to craft a perturbation using typical k-step gradient based method (as used in FGSM, PGD attacks). In other words, they are trying to craft a perturbation that will zero out all other class probability and put maximum weight on the target adversary class when applied to the network.

Experiments were conducted on VGG-Face and Pannous Speech dataset.

1. Paper is easy to follow and different empirical results show a few intuitive observations for a few different settings.

2. The main issue is that the algorithmic is rather simplistic and seems impractical. Most security applications that motivate the paper can easily defend themselves by simply limiting the number of attempts. Even if this core motivation issue is discarded, the algorithmic contribution is very minimal, no analytical understanding either.

3. Some comparison with Black Box models should be added. Although authors claim that Black Box models would require many queries, a specific comparison and contrasting with number of attempts and queries would help understand the critical advantages of the proposed technique.

4. Table 1 provides a good overview of the results but it should include standard deviations for experiments that were averaged.

5. Is there a notion of imperceptibility/attack budget here? Some of the attacks are clearly not imperceptible. This should be discussed.

6. Section 6 seems a little brief and can use more details. Authors claims that using EVM, “the model successfully defeat all our crafted images” but “there is 7.38% chance that the EVM-base model classifies the input as one of the target classes”. This statement is a little confusing.

**Experience Assessment:**

I have published in this field for several years.

**Review Assessment: Checking Correctness Of Derivations And Theory:**

I assessed the sensibility of the derivations and theory.

**Review Assessment: Checking Correctness Of Experiments:**

I assessed the sensibility of the experiments.

**Review Assessment: Thoroughness In Paper Reading:**

I read the paper at least twice and used my best judgement in assessing the paper.

---

> ### Author Response · Authors · 2019-11-08
> **Response to the comments**
>
> 1. We appreciate your comment.
>
> 2. In practice, as you mentioned, real systems have limits on the number of queries in a certain period. That is the main reason most black-box attacks will fail. However, our attack effectiveness is so high that the chance of being successful in the first few attempts is extremely high. For instance, in Table 1, the worst effectiveness is 79% on average. That means that on average each query has a 79% chance of being successful. Hence, it is highly unlikely that setting a threshold, for example 3 attempts, prevents our attack. Note that typical black-box attacks, such Sharif et al. (2016) or Papernot et al. (2017), needs tens to hundreds of queries, if not thousands or more, which is considerably larger than our method and can be easily defended by setting a threshold. Note that we do not query the target model to craft adversarial images. We only use publicly available teacher model to craft images.
> We also use a metric, called Number of Attempts, which may have misled the reviewer, which we will clarify should the paper be accepted. This metric shows the number of attempts needed to trigger all output classes, not just one target class. Our goal is to show that some classes are harder to trigger than others. That number  is not to be compared with the number of queries that black-box attacks need.
> We need to emphasize that our attack rips the benefits of access to the Teacher model. Since the teacher model is publicly available, we assume that the attacker can download the teacher model and craft adversarial attacks with it . As in Algorithm 1, the inner loop (line 6 to 9) works on the local teacher model. We only query the target model in line 10. Hence, most of the processing is done with the local teacher model. That is why we do not need to query the target model as often as black-box attacks.  Note that we only feed the final crafted image to the target model once and if it does not work, we craft another image from scratch by triggering another neuron in activation vector.
> 3. Should the paper be accepted, we will add the comparison to black-box attacks. Note that typical black-box attacks, such Sharif et al. (2016) or Papernot et al. (2017), needs tens to hundreds of queries, if not thousands and more, which is considerably larger than our method and can be easily defended by setting a threshold, as the reviewer suggested. We will release comparison results in the next few days or the revised version, depending on the complexity of running existing black box models.
>
> 4. Thank you for the suggestion. We did not include the standard deviation in Table 1 due to the lack of space. We will revise the manuscript to include it. We did, however, include the confidence interval in Figure 3. In the figure, we can see that the effectiveness of the attack will not be significantly affected in the standard deviation range. For example, if the effectiveness of an average query is reduced from 80% to 60%, the attack is still powerful, and it can bypass the classifier with fewer than 2 attempts on average.
>
> 5. The notion of imperceptibility is not applicable here because the attacker does not have access to any images of the re-training dataset. The crafted images are shown in Figure 5 when different initial images are used. In cases where the starting image is also a (random) image of a person, the crafted image still looks like a person. However, a suspicious human can easily check the input and tell that the input is fake. So, our attack is detectable by a human who knows the identity of persons in each class. Note if the attack can craft images that look like the images of the re-training dataset, it is also an attack on privacy.
>
> 6. We appreciate the reviewer’s comment. Although EVM defeated all our crafted images, it is prone to random face images. We meant that there is 7.38% chance that the EVM-base model classifies a random face images as one of the target classes. The exact sentence is in the paper is: “we find out that by feeding images of random faces (UMass dataset in our study), there is 7:38% chance that the EVM-base model classifies the input as one of the target classes.”

---

> ### Author Response · Authors · 2019-11-15
> **Response to comment 3**
>
> 3. We compare our attack with a black-box attack, called ZOO, in [A], which the source code is available at https://github.com/IBM/ZOO-Attack. We use the same settings as in Section 4.2 of [A] with one exception. We do not stop the optimization process when the target class probability becomes higher than other classes because in our attack scenario, instead, we assume that only images with higher than 95% confidence in one of the classes are accepted. Otherwise, the input is rejected. So, our attack setting is more restricted and that is the reason their black-box attack performs worse in our setting than in their paper. We choose a random image and a random target for the attack. We use the re-trained model on 5 faces as the student model and VGGFace as the teacher model. We set the hard limit of 5,000,000 queries per image. The average query is only computed for successful attacks. In our experiment, the successful rate is 18.2% (in other words, 4 out of 5 attacks fail after 5m attempts). Among the successful ones, Zoo attack needs 1,036,800 queries to the black-box student model on average. In comparison, our attack needs 50,000 queries to the teacher model, 1 query to the black-box student model, and the successful rate (effectiveness) is 91.68%. We need to also emphasize that the number of queries to the teacher model is not critical because the teacher model is publically available and we can download and query the teacher model on a sever of the attacker’s. Additionally, since our attack is target-agnostic, the crafted images only depend on the teacher model. Hence, one can craft a set of potential adversarial images and use it over and over again on multiple student models. It means that, for example, once the crafted images are available, an attacker can use them without a need to query the teacher model. In other words, the number of query is essentially 1 to the student model with an effectiveness of 91.68%. The advantage of using the teacher model gives us enough advantage that no black-box attack without the teacher model can beat us.
>
> Given the time limitation, this is the only model we can implement. We think it clearly demonstrates the difference between our model and a black-box model. If reviewers have specific suggestions on black-box models, we are happy to compare with them further.
>
> [A] Chen, P. Y., Zhang, H., Sharma, Y., Yi, J., & Hsieh, C. J. (2017, November). Zoo: Zeroth order optimization based black-box attacks to deep neural networks without training substitute models. In Proceedings of the 10th ACM Workshop on Artificial Intelligence and Security (pp. 15-26). ACM.

---

### Decision · Program_Chairs · 2019-12-19

**Decision:**

Accept (Poster)

**Comment:**

The reviewers were generally in agreement that the paper presents a valuable contribution and should be accepted for publication. However, I would strongly encourage the authors to carefully read over the reviews and address the suggestions and concerns insofar as possible for the final.